# Hematological Changes in Dogs with Visceral Leishmaniasis Are Associated with Increased IFN-γ and TNF Gene Expression Levels in the Bone Marrow

**DOI:** 10.3390/microorganisms9081618

**Published:** 2021-07-29

**Authors:** Valter Almeida, Isadora Lima, Deborah Fraga, Eugenia Carrillo, Javier Moreno, Washington L. C. dos-Santos

**Affiliations:** 1Fundação Oswaldo Cruz, Centro de Pesquisas Gonçalo Moniz, Salvador 40296-710, BA, Brazil; valter_almeida@live.com (V.A.); isadoraslima@hotmail.com (I.L.); deborah.fraga@ufba.br (D.F.); 2WHO Collaborating Centre for Leishmaniasis, Centro Nacional de Microbiología, Instituto de Salud Carlos III, 28220 Madrid, Spain; ecarrillo@isciii.es (E.C.); javier.moreno@isciii.es (J.M.)

**Keywords:** semidomiciled dogs, hematology, cytokines, bone marrow, *Leishmania infantum*

## Abstract

Visceral leishmaniasis is associated with a variety of hematological abnormalities. In this study, we correlated the hematological changes in the peripheral blood of dogs naturally infected with *Leishmania infantum (L. infantum)* with the distribution of cell lineages and cytokine gene expression patterns in the bone marrow. Samples from 63 naturally semidomiciled dogs living in an endemic area of visceral leishmaniasis were analyzed. *L. infantum* infection was detected in 50 dogs (79.3%). Among those, 18 (32%) had positive splenic cultures and showed more clinical signs. They also had lower red blood cell counts and leukocytosis with an increased number of neutrophils and monocytes in peripheral blood compared to dogs negative to this test. *L. infantum* DNA was detected in the bone marrow of 8/14 dogs with positive splenic culture. Dogs with *L. infantum* infection in the bone marrow presented with histiocytosis (*p* = 0.0046), fewer erythroid cell clusters (*p* = 0.0127) and increased gene expression levels of IFN-γ (*p* = 0.0015) and TNF (*p* = 0.0091). The data shown herein suggest that inflammatory and cytokine gene expression changes in bone marrow may contribute to the peripheral blood hematological changes observed in visceral leishmaniasis.

## 1. Introduction

The zoonotic form of visceral leishmaniasis (VL) is caused by *Leishmania infantum* and is distributed throughout Europe, the Americas, parts of Asia and Africa [1]. Patients with VL normally present with fever, weight loss, hepatosplenomegaly, anemia, leukopenia, thrombocytopenia and hypergammaglobulinemia [2]. Even under treatment, some of these patients develop bleeding and secondary infections and eventually die of the disease [2,3].

Semidomiciled dogs, stray dogs living on the streets with little access to houses, food and water, play a crucial role in maintaining the disease in endemic regions; they are the main domestic reservoir and a source of infection for sand flies, which subsequently transmit the parasites to humans [4,5,6]. Canine visceral leishmaniasis (CanL) has been used as a model for better understanding the disease in humans, due to the similarity of clinical characteristics between them [7,8]. Therefore, the study of CanL is relevant for the understanding of canine and human VL and also to public health.

Dogs with VL show a variety of hematological abnormalities, such as anemia, leukopenia and thrombocytopenia [9]. Some of these changes may be due to enlargement of the spleen and increased hemocateresis. On the other hand, to maintain adequate numbers, blood cells are derived from pluripotent cells present in specialized anatomical microenvironments in the bone marrow. They are reliant on the coordinated interplay between hematopoietic cells and other components of the bone marrow stroma to achieve the fully differentiated stage and arrive in the peripheral blood [10,11,12].

To keep the steady state of hematopoiesis, the bone marrow responds to physiological stresses, such as bleeding or infection, by producing cytokines that can stimulate biological responses in diverse cell types and by keeping the homeostasis in the system [13].

Infections that result in inflammatory changes in bone marrow may interfere with the homeostasis existing between the bone marrow environment and hematopoietic cells due to the immune response altering cytokine gene expression levels. Therefore, changes in the number of mature blood cells may occur in the course of chronic infectious diseases [14].

A high *L. infantum* burden in the bone marrow and spleen is associated with the presence of clinical signs of VL in dogs [15,16,17]. Inflammatory changes leading to disruption of splenic white pulp have been associated with severe VL [18,19,20]. The changes produced by *L. infantum* in those organs are responsible for the most life-threatening manifestations of VL, such as bleeding and susceptibility to bacterial and/or fungal infection [18,19,20].

Cytological studies show that the presence of *L. infantum* amastigotes can interfere with erythroid and thrombocyte lineage differentiation in the bone marrow [14,16,21]. However, the scientific community still needs more studies to explain how those interferences might result in hematological changes seen in patients with VL.

In this study, we contribute to the knowledge of the hematological changes detected in dogs with VL. We correlate the morphological changes and cytokine gene expression patterns in bone marrow with the changes in peripheral blood cell populations in dogs naturally infected with *L. infantum*.

## 2. Materials and Methods

### 2.1. Animals; Samples; and Clinical, Serological and Hematological Analyses

The samples used in this study were obtained from 63 semidomiciled dogs of different breeds and different estimated ages that were collected from the streets of Jequié (Bahia State, Brazil, an endemic area of VL).

The animals were held in a kennel for a period of 5 to 15 days with free access to water and food. Dogs were clinically examined; blood samples were collected from the cephalic vein of the dogs under manual restraint and stored in EDTA-2Na tubes (Greiner Bio-One, Kremsmünster, Austria) and in blood collection tubes (BD Vacutainer) for hematological and serological tests.

General aspects, mucosal conditions, dehydration, lymphadenomegaly, splenomegaly, spleen architecture (histology), onychogryphosis, conjunctivitis, alopecia and other dermatological signs were evaluated and registered. At the end of every analysis, the dogs were included in the asymptomatic category when they had none or only one of the clinical signs described above, oligosymptomatic if they had two or three clinical signs and polysymptomatic when they had more than three clinical signs. The sum of the numbers of clinical signs determined the clinical score of the dogs.

Blood samples for hematological and serum biochemical analyses were collected from the cephalic vein of the dogs under manual restraint. The samples were preserved in EDTA-2Na tubes (Greiner bio-one, Kremsmünster, Austria) and in blood collection tubes (BD Vacutainer; Becton, Dickinson and Co., Franklin Lakes, NJ, USA). Total red blood cell (RBC) and white blood cell (WBC) counts were obtained using an automated cell counter (Pentra 80 counter, ABX Diagnostics, Montpellier, France). Microhematocrit tubes containing the samples were centrifuged at 11,269× *g* for 5 min, and the hematocrit was estimated. Differential blood cell counts were also performed. The serum collected by centrifugation in the Vacutainer tubes was used for the following biochemical tests: total protein, albumin, globulin, aspartate aminotransferase, alanine aminotransferase, blood urea nitrogen and creatinine, using an enzymatic colorimetric method with an A15 auto-analyzer (BioSystems, Barcelona, Spain).

The dogs positive for *L. infantum* infection or not reclaimed by their owners were euthanized and subjected to necropsy. For euthanasia, dogs were sedated with acepromazine (0.1 mg/kg iv, Acepram 1%, Vetnil, São Paulo, Brasil), lethally anesthetized with sodium thiopental (15 mg/kg iv, Thiopentax 1 g, Cristália, São Paulo, Brazil) and euthanized using a saturated solution of potassium chloride (2 mL/kg, iv). Immediately following euthanasia, bone marrow from the sternum or rib was collected for histological and molecular biology analysis. Additionally, splenic aspirates were collected for culture, and fragments of spleen tissue were fixed in formalin and embedded in paraffin for morphological studies.

The presence of anti-*Leishmania* antibodies in the serum was determined by ELISA. Samples for splenic culture were collected by puncturing the spleen using a 20 mL sterile string with an 18 G × 38 mm needle. The splenic aspirates were cultured in a biphasic agar–blood–Schneider medium, supplemented with 10% fetal bovine serum [15,22]. The cultures were examined weekly to identify promastigotes and followed for two months if they remained negative.

### 2.2. Histological Analysis

Samples of the spleen and bone marrow (from the sternum or ribs) were collected during the necropsy of the animals and stored in 10% formalin. Twenty-four hours later, bone samples were transferred to a solution of 5% nitric acid for decalcification. Three-millimeter-thick tissue slices were embedded in paraffin.

Bone tissue sections 4–5 μm in thickness were stained with hematoxylin and eosin and with Giemsa stain and examined by optical microscopy. Bone marrow hyperplasia was defined as a percentage of hematopoietic cells versus fat cells occupying more than 75% of the cavity [23], and bone marrow hypoplasia was defined as a percentage of hematopoietic cells versus fat cells occupying less than 25% of the cavity [24]. To define the cell lineage more affected by hyperplasia or hypoplasia, the myeloid/erythroid cell ratio was calculated by dividing the number of granulocyte cells by the number of nucleated erythroid cells [24]. Histiocytosis was defined as a percentage of macrophages >5% of the bone marrow cells [24]. Erythroid cells are small with round, dense and deeply basophilic nuclei. The erythroid cytoplasm is basophilic in the blast forms with increasing eosinophilia as they mature. Granulocytes have large bean-shaped nuclei that are less basophilic and more vesicular than erythropoietic cells. Megakaryocytes are easily recognized by their large size and multilobulated nuclei.

Disruption of splenic white pulp was defined using the criteria previously described [17]. Briefly, spleen type 1 presents well-organized white pulp allowing a clear distinction of the periarteriolar lymphoid sheath, marginal zone and follicular germinal center and mantle zone. Type 3 spleen presents a disorganized white pulp barely distinct from the other white pulp structures [25].

### 2.3. RNA and DNA Extraction for Molecular Biology

RNA extraction was performed using the SV-Total RNA Isolation System (Promega Corporation, Madison, WI, USA), following the protocol recommended by the manufacturer. All the materials used during molecular biology procedures were free of endonucleases to avoid degradation of the genetic material. The extracted RNA was maintained at −80 °C until use. DNA was obtained from 100 μL of the bone marrow aspirate stored in TRIzol (Thermo Fisher Scientific, Waltham, MA, USA), and was isolated using the phenol–chloroform technique [26]. The obtained DNA was resuspended in sterile distilled water and measured by spectrophotometric reading at 260 nm (Ultrospec 3000, Pharmacia Biotech, Stockholm, Sweden).

### 2.4. Detection of Leishmania infantum Burden in the Bone Marrow

*L. infantum* was detected by PCR specific for the genus *Leishmania*. The target for amplification was the variable region of the gene encoding the RNA of a ribosome subunit. The primers used in the reaction were designed by Van Eys [27] and synthesized by Sigma (Sigma-Genosis, The Woodlands, TX, USA).

The reactions were carried out with Biotools reagents (Biotools B & M Laboratories, Madrid, Spain) and the GenAmp PCR System 2700 thermocycler (PE Applied Biosystems, Foster City, CA, USA). The primers used in the first reaction were R221, which is specific for protozoan kinetoplastids, and R332, which is specific for the genera *Leishmania* and *Crithidia*. For the second reaction, primers R223 and R333, both specific for *Leishmania*, were used. The lengths of the products obtained in the amplification were 603 base pairs (bp) and 353 bp, respectively.

After completion of the first amplification reaction, 25 μL of the obtained product was diluted in 1 mL of sterile distilled water, and 10 μL was subsequently withdrawn from the dilution to proceed with the second PCR. The amplification products of both reactions were visualized by 1.5% agarose gel electrophoresis in TAE buffer (0.04 mM Tris-acetate, 1 mM EDTA, pH 8.0), stained with bioluminescence resonance energy transfer (BRET) (1 μL of a 10 mg/mL solution per 100 mL gel) and using ultraviolet light transillumination (Gel Doc 2000).

### 2.5. Cytokine Gene Amplification by Quantitative (RTqPCR)

Cytokine gene expression in the bone marrow microenvironment was estimated by RTqPCR. Quantification of the gene expression of the cytokines IL-1β, IL-4, IL-10, IFN-γ, TGF-β and TNF was performed using previously described primers and probes [28]. The reactions of reverse transcription (RT) and PCR were performed in the same tube, with the TaqMan PCR Core Reagent Kit (PE Applied Biosystems) and using an ABIPrism 7000 (PE Applied Biosystems). The primers used were synthesized by Sigma (Sigma-Genosys), and the probes were synthesized by Applied Biosystems.

To confirm the efficiency of the technique for detecting IL-4, a positive control composed of concanavalin-stimulated splenocytes was created; however, only this positive control was amplified in the reactions.

### 2.6. Gene Expression Levels and Analysis of the Results

Statistical analyses were carried out using the statistical programs Excel 2016, STATA and GraphPad Plus. The data are shown in tables and graphs. The tests used were the Shapiro–Wilk test to determine the normality of the data distribution, Student’s *t*-test for comparing means of data with normal distribution and Mann–Whitney test for data without normal distribution. Proportions were analyzed using the chi-square test or Fisher’s exact probability test when recommended. The significance level was established at *p* < 0.05.

### 2.7. Ethics Approval

This study was performed in strict accordance with the recommendations of the Brazilian Federal Law on Animal Experimentation (Law 11794) (http://www.planalto.gov.br/ccivil_03/_ato2007-2010/2008/lei/l11794.htm accessed on 1 March 2010) and with the manual for the surveillance and control of VL. The work in the field was conducted in collaboration with the Endemic Diseases Surveillance Program of the State Health Service. All animal manipulations, including euthanasia, were performed under the supervision of at least one veterinarian. The protocol was approved by the Ethics Committee for the Use of Animals in Research (IGM-FIOCRUZ, CEUA, license number: 040/2005).

## 3. Results

### 3.1. General Characteristics of the Dogs

*Leishmania infantum* infection was detected by ELISA or spleen culture in 50/63 (79%) of the dogs. Six culture results were lost due to bacterial infection, and five serology tests were inconclusive and discarded. Fifteen dogs tested positive for both tests, ELISA and splenic culture. The general characteristics of the dogs studied in this work are summarized in Table 1. There was no difference in the sex representation; most animals were mid-sized with estimated ages between two and six years.

### 3.2. Active Infection by L. infantum and Changes in Hematological Parameters

The clinical, hematological and biochemical data of the studied dogs are shown in Table 2. Dogs with positive splenic culture had low red blood cell counts (4.4 ± 1.1), hematocrit (28.2 ± 8.2) and hemoglobin (9.7 ± 2.7) compared to dogs with negative splenic culture (5.2 ± 1.1, 33.8 ± 7.4 and 11.3 ± 2.4, respectively) (Figure 1).

Conversely, dogs with positive splenic culture had higher numbers of white blood cells (19,477 ± 11,863), neutrophils (14,124 ± 10,308) and monocytes (1106 ± 960) than dogs with negative spleen culture (13,554 ± 5405, 8583 ± 4212 and 609 ± 456, respectively) (Figure 2). Those results match the fact that dogs with known active infection proved by splenic culture showed more clinical signs than those with the diagnosis detection by ELISA (Table 2).

### 3.3. Bone Marrow Changes in Dogs with Active L. infantum Infection

Eight out of fourteen (57%) dogs with a positive splenic culture had detectable *L. infantum* DNA in the bone marrow. Among 32 dogs without positive splenic cultures, only 3 had detectable *L. infantum* DNA in the bone marrow (*p* = 0.0004). The histological characteristics of the bone marrow from the dogs studied are shown in Table 2. Bone marrow histiocytosis was observed in 12 dogs (92%) with positive cultures and 16 dogs (47%) with negative splenic cultures (*p* = 0.0046). Increased numbers of lymphocytes, increased numbers of plasma cells and some cases of erythrophagocytosis were also found in the bone marrow of 4, 46 and 13 dogs, respectively.

### 3.4. Hematological Changes in Dogs with Active Infection and Splenic White Pulp Disorganization

Since splenic white pulp disruption is associated with severe forms of VL (18), we studied the hematological changes in dogs with this spleen histological characteristic. Dogs with active infection and structural disorganization of the spleen had lower red blood cell counts (3.5 ± 1.1) and more events of erythroid hypoplasia compared with the dogs without active infection and with normal splenic histology (5.3 ± 1.2) (*p* = 0.0066) (Figure 3).

Furthermore, there was a significant increase in the white blood cell count (30,060 ± 17,030) with neutrophilia (23,881 ± 12,994) in dogs with a positive splenic culture and disrupted white pulp when compared with dogs without active infection and with normal spleen (13,800 ± 2173 and 7548 ± 2988; *p* = 0.0064 and *p* = 0.0015, respectively), (Figure 4).

### 3.5. Cytokine Gene Expression Levels in Bone Marrow and Hematological Changes in Dogs Infected with L. infantum

Among the 46 samples of dogs that had bone marrow examined by nested PCR for *L. infantum* DNA detection, 11 (24%) were positive. The dogs with detectable *L. infantum* in the bone marrow had a significant decrease in the number of circulating red blood cells (*p* = 0.0030) and hemoglobin concentrations in the blood (*p* = 0.0021). Furthermore, these animals presented higher gene expression levels of IFN-γ (*p* = 0.0015) and TNF (*p* = 0.0091) than animals with negative PCRs for parasite DNA detection in the bone marrow (Figure 5).

The presence of *L. infantum* in the bone marrow did not change the gene expression levels of TGF-β, IL-1β or IL-10 in the bone marrow of the dogs in this study. IL-4 was not detected in the bone marrow collected from the dogs analyzed in this study.

## 4. Discussion

In this work, we studied the changes in the peripheral blood cells and bone marrow of dogs naturally infected with *L. infantum* living in semidomiciled conditions in endemic areas in Brazil. We show that animals with active *L. infantum* infection had anemia, leukocytosis with neutrophilia and an increase in the gene expression levels of IFN-γ and TNF in the bone marrow. The animals with the most severe forms of the disease with disruption of splenic white pulp also had erythroid hypoplasia.

The reduction in circulating red blood cells is a typical laboratory sign of VL [15,29,30,31]. Although it has been known and reported over the years [29,32,33,34], the severity and pathogenesis of anemia in VL are still not fully comprehended [35]. Hypersplenism has been considered the main cause of the pancytopenia found in VL. In humans with relapsing forms of VL, splenectomy leads to an increase in all blood cell counts [36]. Trapping and removal of blood cells from the circulation by an enlarged spleen is a potential mechanism of pancytopenia. However, in this work, we also show hypoplasia of the erythroblastic series in the bone marrow of animals with severe VL. Nicolato et al. [16] found similar results in myelograms of symptomatic dogs with VL. This finding suggests a potential role for bone marrow dysfunction induced by the *L. infantum* infection.

The dogs with active *L. infantum* infections in which the parasite DNA was detected in bone marrow had a significant increase in IFN-γ and TNF gene expression. These cytokines are widely expressed in many tissues in VL [37]. They exert many inflammatory actions in the disease. There are several mechanisms by which IFN-γ can negatively influence the number of red blood cells.

IFN-γ decreases hematopoietic activity in bone marrow [38] and considerably affects the differentiation of most hematopoietic progenitor cells [39]. In chronic diseases, IFN-γ affects iron metabolism and limits iron during erythropoiesis [40]. Additionally, IFN-γ increases the number of activated macrophages that can promote erythrophagocytosis, contributing to the decrease in circulating erythrocytes [41].

It has been proposed that IFN-γ induces the transcription factor PU.1, resulting in the inhibition of GATA-1 and consequent interference in the differentiation of erythroid cells [41,42]. TNF has inhibitory effects on hematopoietic stem cells in mice [43], as well as on human erythroid progenitors (BFU-E and CFU-E) [44]. Furthermore, erythroid and megakaryocytic dysplasia are associated with increased production of IFN-γ and TNF in the bone marrow of dogs infected with *L. infantum* [14].

The histiocytosis found in the bone marrow of dogs with active infection and detection of *L. infantum* may reflect an impairment of bone marrow homeostasis in dogs with VL. Therefore, the increase in IFN-γ and TNF gene expression levels observed in these animals may negatively influence the differentiation of the erythroid lineage in bone marrow, aggravating the anemia presented by dogs with active *L. infantum* infections.

Low white blood cell counts have been described in canine and human VL [16,31,36]. In this study, however, we found an increase in blood leukocytes. Leukocytosis with neutrophilia has been reported in dogs with VL and is considered a common hematological disturbance in symptomatic cases [19,45,46].

Semidomiciled dogs in endemic areas may be reinfected by *L. infantum* and infected by other pathogens that may stimulate the leukocyte response. For instance, bacterial conjunctivitis and skin ulcers are frequently found in animals living in similar conditions [19,47]. It is also interesting to note that the increase in blood leukocytes was even higher in animals with white pulp disruption.

The spleen plays a central role in the protection against bacterial infection. Therefore, the disorganization of the splenic compartments caused by the *L. infantum* infection may also increase susceptibility to infection [48]. Further studies are necessary to confirm whether the inflammatory state induced by bacteria coinfections is responsible for continuous stimulation of leukocyte release into the blood resulting in leukocytosis.

In dogs, megakaryocytic dysplasia, erythrophagocytosis, erythroid dysplasia and emperipolesis have been related to and associated with natural infection by *L. infantum* [14]. An increase in the percentage of lymphocytes and plasma cells, erythroid hypoplasia and erythrophagocytosis have been found in dogs with VL [49]. With the exception of erythroid dysplasia, similar changes were found in our study; however, the difference between groups was not statistically significant.

This study aimed to associate hematological disorders with the distribution of cell lineages and the pattern of cytokine gene expression in the bone marrow of dogs naturally infected with *L. infantum*. Currently, few studies have related the disruptions of bone marrow microenvironments with changes in the distribution pattern of cell populations in the blood in VL.

## 5. Conclusions

Dogs with severe active *L. infantum* infection have anemia and leukocytosis with neutrophilia. These peripheral blood changes are accompanied by erythroid hypoplasia in the bone marrow. The hematological alterations presented by dogs with VL are more profound when there is splenic disorganization and are associated with changes in the pattern of IFN-γ and TNF gene expression levels in the bone marrow.

## Figures and Tables

**Figure 1 microorganisms-09-01618-f001:**
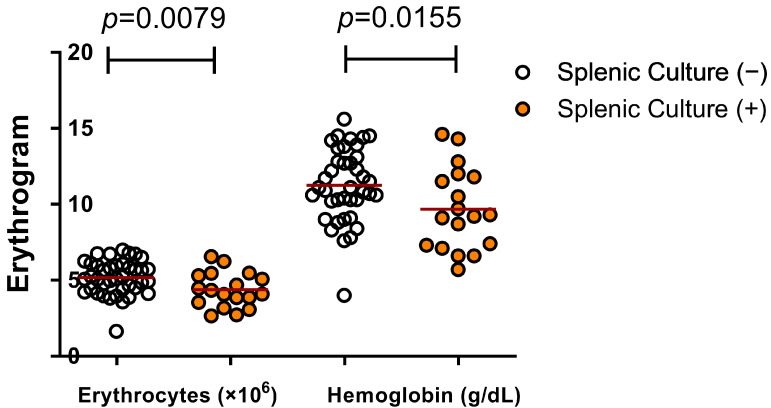
Hematological disorders presented by dogs with positive spleen culture indicating active infection by *L. infantum* (Student’s *t*-test).

**Figure 2 microorganisms-09-01618-f002:**
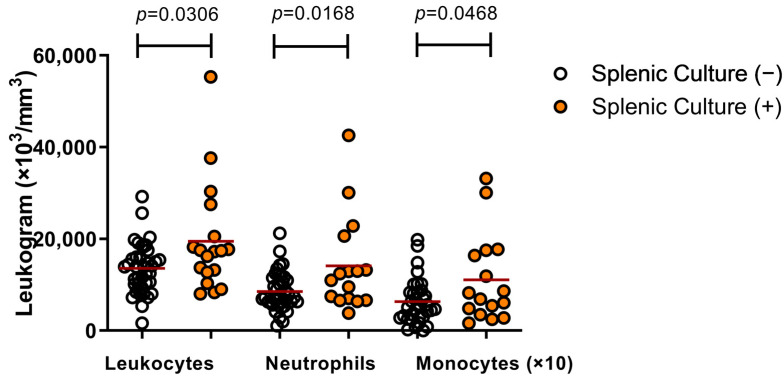
Changes in leukogram presented by dogs with active infection by *L. infantum* (Mann–Whitney).

**Figure 3 microorganisms-09-01618-f003:**
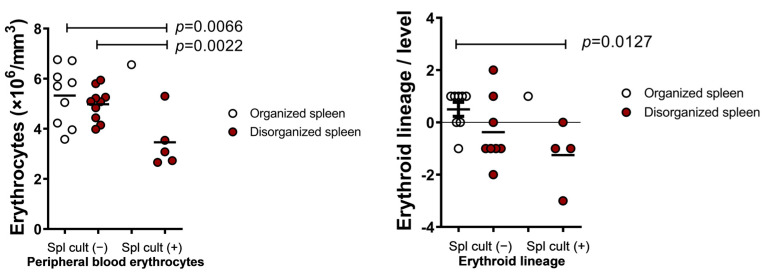
Hematological alterations presented by dogs with active infection of *L. infantum* associated with splenic disorganization. Spl cult (−) = negative splenic culture; Spl cult (+) = positive splenic culture. (Mann–Whitney and Student’s *t*-test.)

**Figure 4 microorganisms-09-01618-f004:**
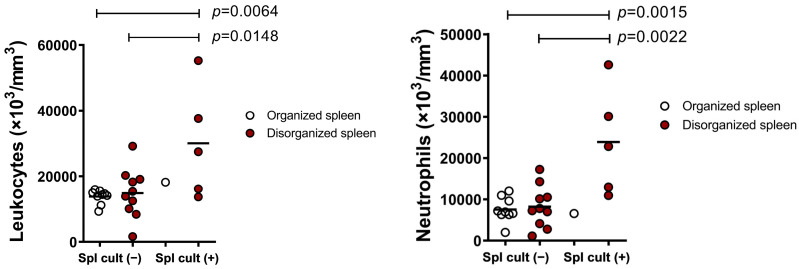
Changes in leukogram of dogs with active infection of *L. infantum* and splenic disorganization. Spl cult (−) = negative splenic culture and Spl cult (+) = positive splenic culture. (Student’s *t*-test.)

**Figure 5 microorganisms-09-01618-f005:**
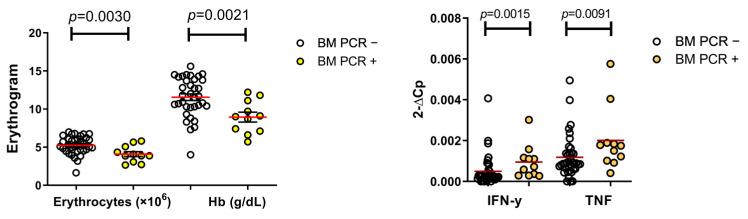
Erythrocytes, hemoglobin and gene expression levels of IFN-γ and TNF in the bone marrow of infected dogs. Hb = hemoglobin; BM = bone marrow. (Mann–Whitney and Student’s *t*-test.)

**Table 1 microorganisms-09-01618-t001:** General characteristics of semidomiciled dogs collected from the streets of Jequié, BA, Brazil, in 2010.

Parameter	Frequency	(%)
N	63	100
Male	34/63	54.0
Female	29/63	46.0
Size:		
Small	19/63	30.2
Middle	38/63	60.3
Large	06/63	9.5
Estimated age:		
Six months to 2 years old	17/63	27.0
2–6 years old	38/63	60.3
More than 6 years old	08/63	12.7
Detection of *L. infantum* infection:		
Infection confirmed by any test (total)	50/63	79.3
Spleen culture	18/57	31.6
ELISA	47/58	81.0
Clinical category:		
Oligosymptomatic (2 or 3 clinical signs)	24/63	38.1
Polysymptomatic (more than 3 clinical signs)	39/63	61.9

**Table 2 microorganisms-09-01618-t002:** General results of the different tests detecting *L. infantum* infection in semidomiciled dogs.

Normal Parameters	ELISA (*n* = 47)	Spleen Culture (*n* = 18)
Clinical score	3.7 **^a^**—SD: 1.4	5.1 **^b^**—SD: 1.4
Blood tests		
Erythrogram (5.5–8.5 × 10^6^/μL)	5.3 **^c^**—SD: 1.1	4.4 **^d^**—SD: 1.1
Hemoglobin (12–18 g/dL)	11.3 **^e^**—SD: 2.4	9.7 **^f^**—SD: 2.7
Leukocytes (6–17 × 10^3^/μL)	13050 **^g^**—SD: 4637	19478 **^h^**—SD: 11863
Platelets (200,000–500,000/mm)	132521—SD: 113142	148588—SD: 120859
Urea (21.4–59.92 mg/dL)	40.2—SD: 16.6	43.8—SD: 16.3
Creatinine (0.5–1.5 mg/dL)	0.82—SD: 0.25	0.71—SD: 0.27
Total proteins (5.4–7.1 g/dL)	10.9—SD: 1.5	11.0—SD: 2.6
Albumin (2.6–3.3 g/dL)	3.0—SD: 0.8	2.7—SD: 0.9
Globulin (2.7-4.4 g/dL)	7.9—SD: 1.4	8.0—SD: 1.7
A/G ratio	0.4—SD: 0.1	0.3—SD: 0.1
AST (21–45 UI/L)	37.0 **^k^**—SD: 24.7	63.0 **^l^**—SD: 55.0
ALT (21–73 UI/L)	37.7—SD: 17.1	46.8—SD: 24.3
TGL (20–112 mg/dL)	51.5 **^m^**—SD: 25.8	70.6 **^n^**—SD: 22.5
Bone marrow histology	(*n* = 26)	(*n* = 13)
Cellularity	90.0%—SD: 11.7	94.2%—SD: 4.9
Erythroid hypoplasia	35%—N = 09/26	23%—N = 03/13
Granulocytic hyperplasia	96%—N = 25/26	100%—N = 13/13
Megak. hyperplasia	73%—N = 19/26	54%—N = 07/13
Histiocytosis	47% **^i^**—N = 12/26	92% **^j^**—N = 11/13
Hemorrhage	35%—N = 09/26	15%—N = 02/13
Congestion	27%—N = 07/26	31%—N = 05/13
Emperipolesis	35%—N = 09/26	23%—N = 03/13
Megakaryocyte dysplasia	69%—N = 18/26	39%—N = 05/13

The values demonstrated are the mean of the results and the number of cases. The number of samples used to analyze the bone marrow histology differs from the group due to the bone marrow sample quality or processing issues. SD = standard deviation. Please note *p*-value comparisons on the same lines: **^a^**^,**b**^ *p* = 0.0015; **^c^**^,**d**^ *p* = 0.0063; **^e^**^,**f**^ *p* = 0.0186; **^g^**^,**h**^ *p* = 0.0232; **^i^**^,**j**^ *p* = 0.0046; **^k^**^,**l**^ *p* = 0.0096, **^m^**^,**n**^ *p* = 0.0046.

## Data Availability

The data presented in this study are openly available in FigShare at DOI reference number: 10.6084/m9.figshare.14776494.

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
