# Peer review of "Hematological Changes in Dogs with Visceral Leishmaniasis Are Associated with Increased IFN-γ and TNF Gene Expression Levels in the Bone Marrow"

_microorganisms, 2021, doi:10.3390/microorganisms9081618_

Round 1
Reviewer 1 Report
The manuscript by Almeida et al. describes an observational study performed in dogs with canine leishmaniasis. The study evaluates the haematological changes possible associated with Leishmania infantum infection in peripheral blood, spleen, and bone marrow and establishes relations with the detection of Leishmania parasites and the gene expression of pro and anti-inflammatory cytokines. The authors concluded that the generation of IFN-gamma and TNF by bone marrow are related to the disruption of spleen compartments and the haematological changes observed in proved infected dogs. Despite the study interest, the manuscript presents some inconsistencies that need to be taken into consideration by the authors.
Specific comments and suggestions:
- Title – This study examined the accumulation of cytokine mRNA in the bone marrow. Therefore, the cytokine gene expression has been evaluated, which is different from cytokine production (expression). A direct correlation between cytokine gene expression and cytokine production is not observed in all cases. Thus, gene expression (an indirect indication of possible cytokine generation) should be applied in the title and throughout the manuscript and correctly used in the Discussion section (line 317).
2. In the abstract and throughout the manuscript should be corrected the spelling of leukocytosis and leukocytes.
- Introduction
3.1. Line 34 – The concept of semi-domiciled dogs needs to be explained, including the difference between semi-domiciled dogs and domestic and feral dogs. It also should be indicated the importance of the semi-domiciled dogs for the present study. If this study has been done with domestic dogs, do you expect different results? Furthermore, the concept of semi-domesticated dogs (Discussion section, line 324) also should be defined.
3.2. Line 54 - Since immune mediators, such as cytokines, make part of an immune response, please explain what do you mean with the following sentence “…. the immune response altering cytokine expression.”
3.3. Line 56 – Replace parasite by Leishmania at the beginning of the sentence.
3.4. Lines 64-65 - This statement is of difficult understanding and needs to be improved
3.5. Line 69 – To improve clarity and facilitate understanding, patients can be replaced by dogs
- Material and Methods
4.1. Lines 86-87 – The methodology used to define the clinical scores should be briefly described.
4.2. Lines 76-78 and lines 101-102 – These statements seem contradictory and need to be clarified.
4.3. Line 95 - rpm should be converted into x g.
4.4. Lines 110-111 - The methodology used in the serological assays and splenic cultures can be briefly described.
- Results
5.1 Line 195 - For a better understanding of results, please specify which methodology was used to detect 81% of infection. This means that most semi-domiciled dogs can be infected by Leishmania. Is this an expected value?
5.2. Line 207 – Active infection is indicated in subheading 3.2 and then several times throughout the manuscript. Please explain what you mean by active infection. It is when dogs present positive spleen cultures? On the other hand, it is also referred “without active infection” (line 258). Does this mean that these dogs had no detectable parasites?
5.3. Lines 211-212 - This sentence needs improvement.
5.4. Line 260 – Please improve the sentence.
5.5. Table 2 – The legend needs to be improved. For example, what represents the value of 39.8 of urea? Is it a median (or mean) of urea value of dogs that had a positive PCR?
5.6. Figures – Despite we can find statistical tests listed in the Material and Methods section is important to specify in figure legends the statistical methods used in each case. Moreover, the meaning of figure abbreviations should be indicated in the legends.
5.7. It would be nice to have some representative histopathology images to illustrate the described results.
- Discussion
6.1. Lines 329 – What do you mean when refers that the susceptibility to infection increases in consequence of disorganization of splenic compartments? Do you hypothesize that the dogs already had spleen changes due to other (bacterial) infections? Or is the presence of Leishmania parasites that lead to spleen alterations? Please, clarify the statement.
Reviewer 2 Report
This study investigates the haematological deviations and immunological responses linked to canine VL. The study is well conducted and provides an interesting contribution to the field. I do have some comments/questions regarding the presentation of the results and the design of the study, which the authors might choose to clarify.
Primary comments:
Design:
The study did not include negative controls. Could the authors comment on what is the reason for this? (If for ethics reasons: what about the relatively noninvasive procedures like taking a blood sample?)
Results:
- Table 1: "Detection of L. infantum infection". The authors should clarify why the sample sizes between the different tests (Spleen Culture / ELISA / PCR) differ. They explain that 5 cultures failed due to bacterial infection for the Spleen Culture, however, this brings the Spleen total to 62 and not 63. For ELISA and PCR, no explanation is given. It would have been interesting to know if all 63 dogs were PCR positive.
Table 2:
-Units are missing in this table. Please add them.
-Also reference value ranges for healthy individuals would be interesting to add.
- It is unclear to which comparisons the p-values refer.
Minor Comments:
L42: The authors mean 'hemocateresis'
L193: Were all collected dogs included or only those that had Leishmaniasis symptoms? This was not clear to me while reading the manuscript. Also in the abstract, this is unclear.
Round 2
Reviewer 1 Report
The revised manuscript shows a considerable improvement, and it is my opinion that can be accepted for publication. However, the following minor corrections are suggested:
Line 228-229 - The use of parentheses used indicate red blood cells, haematocrit and haemoglobin values (5.2 ± 1.1, 33.8 ± 7.4, and 11.3 ± 2.4, respectively) of dogs with negative spleen cultures can be reduced;
Lines 230, 234 and 289 - Add parenthesis to Figures;
Table 2 - Leukocyte correction is missing.
